# Study of the Impact of Data Compression on the Energy Consumption Required for Data Transmission in a Microcontroller-Based System

**DOI:** 10.3390/s24010224

**Published:** 2023-12-30

**Authors:** Dominik Piątkowski, Tobiasz Puślecki, Krzysztof Walkowiak

**Affiliations:** Faculty of Information and Communication Technology, Wrocław University of Science and Technology, 50-370 Wrocław, Polandtobiasz.puslecki@pwr.edu.pl (T.P.)

**Keywords:** embedded systems, data compression, data transmission, Huffman, LZ77, LZ78, LZW, JPEG, TinyML, Internet of Things

## Abstract

As the number of Internet of Things (IoT) devices continues to rise dramatically each day, the data generated and transmitted by them follow similar trends. Given that a significant portion of these embedded devices operate on battery power, energy conservation becomes a crucial factor in their design. This paper aims to investigate the impact of data compression on the energy consumption required for data transmission. To achieve this goal, we conduct a comprehensive study using various transmission modules in a severely resource-limited microcontroller-based system designed for battery power. Our study evaluates the performance of several compression algorithms, conducting a detailed analysis of computational and memory complexity, along with performance metrics. The primary finding of our study is that by carefully selecting an algorithm for compressing different types of data before transmission, a significant amount of energy can be saved. Moreover, our investigation demonstrates that for a battery-powered embedded device transmitting sensor data based on the STM32F411CE microcontroller, the recommended transmission module is the nRF24L01+ board, as it requires the least amount of energy to transmit one byte of data. This module is most effective when combined with the LZ78 algorithm for optimal energy and time efficiency. In the case of image data, our findings indicate that the use of the JPEG algorithm for compression yields the best results. Overall, our research underscores the importance of selecting appropriate compression algorithms tailored to specific data types, contributing to enhanced energy efficiency in IoT devices.

## 1. Introduction

There are numerous embedded devices present in various aspects of our everyday lives, many of which are equipped with some form of connectivity, effectively classifying them as Internet of Things (IoT) devices. As indicated by Marjani et al. [1], the number of sensors is projected to increase by 1 trillion in 2030, directly impacting the volume of generated and transmitted data. The escalating data production from the ever-growing array of IoT devices introduces the concept of Big Data into the realm of IoT [2].

Given the non-negligible prevalence of battery-powered embedded systems, the aspect of energy conservation becomes crucial. An intriguing recent concept related to embedded devices is TinyML (Tiny Machine Learning) [3,4,5]. If the collected data were analyzed by a Machine Learning (ML) model, only time-critical data could be promptly transmitted for immediate analysis, obviating the need for frequent, relatively large bidirectional data transmissions. This does not imply that non-critical data should be discarded; instead, data could be buffered on the device, compressed, and sent as a complete buffer later. This approach provides diagnostic data for more sophisticated cloud-based models, which could predict device malfunctions or assess machine conditions. Additionally, the gathered data can serve as further training material for ML models. In the case of the presented weather data example, it would take almost 40 h to exhaust the device memory, eliminating the need for additional data storage as data can be transmitted when the buffer reaches its capacity. It is noteworthy that the existing wireless communication board can be utilized, resulting in no additional hardware requirements. To the best of our knowledge, there is no TinyML research paper evaluating the impact of data compression on the energy consumption required for data transmission in a microcontroller-based system with resources comparable to the used STM32F411CE chip.

The utilization of TinyML has a positive impact on energy consumption, as the energy requirement of microcontroller-based devices remains low even at maximum workload levels [4], and the energy required for transmission is significantly greater than that needed for TinyML-related computations [5].

The motivation behind this paper is linked to the minimization of energy consumption in battery-powered embedded devices. Specifically, our focus is on identifying the most suitable method for data processing that results in the lowest energy requirement for data transmission. This not only reduces the wear on rechargeable batteries, extending each charge cycle and prolonging battery lifespan but also delays the replacement and disposal of rechargeable batteries. In the case of non-rechargeable batteries, this becomes even more crucial, as a discharged battery must be replaced and properly disposed of after a single use. Therefore, optimizing energy consumption can significantly reduce e-waste production. Additionally, transmitting compressed data has the secondary benefit of putting less strain on the utilized network compared to uncompressed data transmission, minimizing the transmitted amount of data.

The main hypothesis of our paper is as follows: it is possible to lower the energy consumption required for data transmission in a microcontroller-based system by compressing the data using carefully selected algorithms. The primary challenge in this research is the power consumption constraint of the chosen microcontroller, which directly influences the amount of available resources, as low-power, efficient microcontrollers have very limited memory and computational power. To address this, various transmission modules need to be tested on a test bench with a microcontroller in scenarios involving the transmission of various uncompressed data, as well as compressing the data and sending it afterward to evaluate the impact of data compression on the energy consumption required for data transmission. The main contribution of this paper is the study of the impact of data compression, using various compression algorithms, on the energy consumption required for data transmission, utilizing various transmission modules in a severely resource-limited microcontroller-based system intended to be battery-powered.

This paper is organized as follows: In Section 2, related works are discussed, emphasizing the discovered research gap. In Section 3, the utilized hardware, i.e., the chosen microcontroller and transmission modules, as well as data and algorithms, are described. Limitations of the chosen microcontroller are discussed, followed by the method of compressible data generation used to benchmark the approaches. The operating principle of each implemented and tested algorithm is described, along with implementation details that directly influence the resulting compressed data size and enable the algorithms to run on severely constrained microcontroller hardware. Details and caveats of each transmission module are discussed, followed by important measurement details. In Section 4, the results are presented, including time and energy ratios, as well as measurements of each compression algorithm and transmission module. Section 5 contains a discussion of the results. Section 6 concludes this paper.

## 2. Related Works

Encountering an embedded, microcontroller-based system equipped with Machine Learning (ML) to transform it into a TinyML device capable of prediction, classification, and decision-making is not uncommon. Numerous TinyML use cases share a commonality—a significant amount of data. For instance, an electric motor anomaly detection system, as detailed in [6], could be implemented using a TinyML device. This device would conduct data harvesting and analysis on-board, transmitting detected anomalies promptly. Valuable non-anomalous data can be utilized for diagnostics to evaluate potential device malfunctions, assess machine conditions, and serve as additional training material for ML models. Given that this data does not require critical priority transmission, it can be buffered on the device, compressed, and sent as a complete buffer later to conserve energy. This is particularly crucial for battery-powered industrial sensors, where routing additional wires might lead to OSHA violations due to the risk of wires becoming entangled around the motor shaft. Minimizing energy consumption is imperative to prolong the lifespan of battery-powered TinyML devices, driven by the following factors:Longer battery life in these systems minimizes maintenance costs by reducing the frequency of battery replacements.These systems may be situated in hard-to-reach locations, and replacing batteries might require the serviced machine to halt its operation, resulting in significant downtime during each battery replacement.

There are other papers that analyze compression in the context of TinyML. An example can be found in [7], where a new data compression algorithm tailored for TinyML is proposed. As stated by researchers in [3], Huffman encoding is the most popular encoding method in ML. This implies that an evaluation of results produced by the Huffman algorithm can provide a valuable baseline for less commonly used methods. To the best of our knowledge, the Lempel–Ziv algorithm family is not widely used in TinyML. In total, four papers mentioning TinyML with any variation of Lempel–Ziv were found at the time of this research. Two of these works acknowledge the existence of these algorithms but employ a different approach [8,9]. One paper [10] refers to [11], presenting an approach that uses Lempel–Ziv–Welch as one of the compression steps. There is also a highly specialized paper, with a signal processing use case, that uses Lempel–Ziv–Welch to encode data [12].

Since there are also numerous TinyML image-driven use cases, such as fast image recognition with a low memory footprint [13], image data transmission is not an implausible scenario. This implies that the study of the impact of image data compression on energy consumption required for data transmission in a microcontroller-based system is important and cannot be omitted.

The previous papers [10,11,14,15,16,17,18,19] on this topic have primarily focused on compression algorithms such as LZW and its variations (e.g., s-LZW) and have often employed hardware with substantial computing power, such as the Raspberry Pi 4B with 8 GB RAM. In contrast, this paper introduces, as the main contribution and novelty of our work, the following new elements: the evaluation of canonical Huffman, Lempel–Ziv 77, Lempel–Ziv 78, and Lempel–Ziv–Welch algorithms in terms of energy and time efficiency for compressing and transmitting text data and the evaluation of the Joint Photographic Experts Group algorithm in terms of energy and time efficiency for compressing and transmitting image data. All evaluations were conducted using the STM32F411CE microcontroller, which is suitable for building TinyML devices due to its low power consumption (100 μA/MHz core current), a 100 MHz ARM Cortex M4 core providing ample computational power, and a decent-for-TinyML-purposes 512 KB of Flash and 128 KB of RAM. It is important to note that many of the evaluated algorithms were not used or compared to previous papers.

There are also existing research papers [20,21,22,23] that explore the topics of TinyML and IoT in a manner similar to this paper.

Moreover, to the best of our knowledge, there is no TinyML research paper that evaluates the impact of data compression on the energy consumption required for data transmission in a microcontroller-based system with resources comparable to the used STM32F411CE chip. Therefore, to fill this research gap, in the following sections of this paper, we present a study of the impact of data compression on the energy consumption required for data transmission in a microcontroller-based system.

## 3. Materials and Methods

This section presents the methods used in the proposed test bench, combined with their implementation details, as well as the sources of data used to benchmark the compression algorithms. There are various compression algorithms, each with its own benefits and drawbacks. A selection of simple, well-known algorithms was chosen because of the relatively low amount of resources available on the chosen embedded platform. The presence of important requirements of minimizing time and energy consumption can be satisfied in an easier way by utilizing lightweight methods.

### 3.1. Hardware

Embedded systems are often battery-powered. This introduces an important factor to be taken into consideration—energy consumption. In order to prolong the battery life of such a system, evaluation of the energy efficiency of every part of the design is mandatory. This includes the heart of every embedded system—a microcontroller. In this paper, the Blackpill board with the STM32F411CE microcontroller [24] was chosen due to several factors:Availability: a very popular board.Price: about $10.Performance: 100 MHz ARM Cortex M4.Memory size: 512 KB of Flash and 128 KB of RAM.Efficiency: 100 μA/MHz core current consumption.

The Blackpill board has a Light Emitting Diode (LED) indicating power status. As constantly powered LED is not wanted in battery-powered, energy-efficient design, it is reasonable to disconnect it on the measured board.

Three transmission modules are analyzed in our paper: FS1000A, nRF24L01+, and ESP-01.

FS1000A [25] is the cheapest of all tested transmission modules, available for under $2. The detailed specification of this module is presented in Table 1. Due to the fact that the 433 MHz band is one of the most popular choices for wireless devices in Europe, and also that the ASK modulation is sensitive to various interference types, adequate transmission protocol is required for successful transmission. Another important aspect that must be taken into consideration is the requirement of priming the Automatic Gain Control (AGC) of the receiver by supplying a long enough signal at the beginning of transmission. The used transmission protocol is shown on the Listing 1.

**Listing 1.** Protocol for FS1000A.It should be noted that a logical one in ASK modulation implemented on this board is equivalent to signal presence, and a logical zero is equivalent to no signal presence.2000 μs of logical one500 μs of logical zeroData encoded with Manchester II [26]:–Low-high transition for bit 0.–High-low transition for bit 1.1000 μs of logical one, indicating the end of transmission

Requirement of priming the AGC of the receiver is satisfied with 2000 μs of signal presence at the beginning of transmission. Utilization of Manchester II [26] code solves two problems:Signal is present for 50% of the time during data transmission and is frequently alternating its state, solving the problem of keeping the receiver’s AGC at the right level.It produces a self-clocking signal, solving the problem of receiver synchronization.

Manchester II code comes at a cost of double the required bandwidth, as sending the same bit twice requires the signal to also change at the beginning of bit transmission. This results in reduced transmission bit rate—speeds of up to 5 kbps can be achieved without errors, but 10 kbps transmission is affected by severe deterioration of signal integrity. Measurements of transmission speed are shown in Figure A1.

As neither the transmitter nor receiver board has an onboard antenna, it is mandatory to add one before conducting transmission tests. The electromagnetic radiation wavelength can be calculated using the Formula (Equation 1).
(1)λ=c/f,
where λ—wavelength, *c*—speed of light and *f*—frequency.

For 433 MHz, the wavelength is equal to 69.236 cm. As a full-wave antenna of this size is not realistically usable in an embedded device, another approach is required. It is not uncommon to see half-wave and quarter-wave antennas. In fact, quarter-wave monopole is one of the most common antennas found in portable devices [27]. For 433 MHz, the half-wave antenna size is equal to 34.618 cm, and the quarter-wave antenna size is equal to 17.309 cm. Due to the size acceptable for the embedded device, as well as a suitable radiation pattern, a quarter-wave monopole antenna design was chosen.

nRF24L01+ module based on nRF24L01+ chip [28] is an ultra-low power transceiver available for about $2. The detailed specification of this module is reported in Table 1. This module automatically handles data packets, requiring no additional protocol specification, which was mandatory for FS1000A. It also supports dynamic payload size, automatic hardware acknowledgment (ACK), automatic packet integrity checking with Cyclic Redundancy Check (CRC) and receiving data from six devices in parallel. As the nRF24L01+ module comes with a microstrip antenna, there is no need for an external one.

ESP-01 is a transceiver module based on ESP8266EX chip [29] available for under $3. The detailed specification is shown in Table 1. This module, like nRF24L01+, comes with a microstrip antenna, and there is no need for an external one. ESP-01 module has two LEDs, indicating power status and UART transmission. As these LEDs are not needed for module operation, they were disconnected.

### 3.2. Data

In order to conduct research on compression algorithms, it is required to gather compressible data. Some of the algorithms can be specialized in particular data types, for example, Joint Photographic Experts Group (JPEG) algorithm [30] is used for images. This creates a need for compressible data of various types in order to measure the results correctly.

#### 3.2.1. Weather

In order to generate compressible text data, a simple weather station was created. It consists of Bluepill board with STM32F103CB microcontroller and BME280 sensor board.

BME280 is a low-power sensor, capable of measuring temperature, pressure, and relative humidity [31]. Due to the small package size (8-pin LGA, 2.5 × 2.5 mm), it is often found on breakout boards with a 2.54 mm pitch goldpin connector. BME280 measurement ranges are shown in Table 2.

Measurements were conducted every minute for 24 h with an oversampling factor of 16. The weather data format is shown on the Listing 2, and each section is described in Table 3. It should be noted that leading zeros are omitted in temperature, pressure and relative humidity sections.

**Listing 2.** Weather data format.
YYYY-MM-DD HH:mm NNNNNNNN T: tt.tt P: pppp.pp H: hh.hh\r\n


#### 3.2.2. Images

A popular camera module used in embedded devices is based on OV7670 [32]. Due to the fact, that it is difficult to accurately measure the time and energy required for compression of RGB888 image with size of 640 by 480 px, as it would require 900 KB of memory to store uncompressed image while the Blackpill board has only 128 KB of RAM and 512 KB of Flash memory, another approach was required, as only smaller image sizes would fit in the Blackpill board memory—320 by 240 px image with size of 225 KB could fit in Flash memory for benchmarking purposes, and 160 by 120 px image with size of 56.25 KB could fit either in Flash or RAM memory. As measuring the compression of images from the OV7670 module would include data acquisition, synthetic data were used instead. Tested images consist of rows generated with pseudocode shown on the Listing 3.

**Listing 3.** Row generation pseudocode.
uint16_t size = image_width * 3;

uint_8 row[size];

for (uint16_t i = 0; i < size; i++)

    row[i] = i % 256;


It should be noted that image width is multiplied by 3 due to the fact that RGB888 format uses 3 bytes per pixel.

### 3.3. Compression Algorithms

#### 3.3.1. Huffman

Huffman coding [33] is one of the simplest methods of lossless data compression. Due to the nondeterministic result of the constructed Huffman tree—storing the dictionary requires storing both code words and symbols as well as code length. This inconvenience can be solved by utilizing canonical Huffman code. It should be noted that code word lengths will not change.

By using canonical Huffman code, the dictionary can be represented as:Size of dictionary.Number of symbols with code word length of 1…n, where *n* is the longest code word length.Symbols are sorted by code word length, and then by the symbol.

There is still room for improvement, this time in terms of efficient encoding of these values. Considering there will never exist a dictionary with a size less than 1 and greater than 256, it can be stored as a size decremented by one, effectively utilizing the 0…255 range that perfectly fits in one byte. The number of symbols with a given code word length can be safely stored as one byte each, as it will never reach 255. Symbols by design need to be stored as one byte each. This approach is more efficient in comparison with the non-canonical Huffman dictionary, which requires:ASCII symbol.Code word.Code word length.for each entry, as well as dictionary size.

In order to make the data decompressible without additional noise at the end caused by trailing zeros that could be interpreted as valid symbols, it is a good idea to include data size in the compressed data header. As STM32F411CE has 128 KB of RAM, a 16-bit compressed data size should be sufficient.

#### 3.3.2. Lempel–Ziv 77

Lempel–Ziv 77 (LZ77) [34] is a lossless data compression algorithm that leverages the existence of repeating strings of symbols in data to compress it.

In terms of implementation, the buffer can be implemented as a circular buffer in order to avoid time-costly memory copy operations. It should be noted that due to the dictionary’s dependency on previously processed symbols, there is no need to store any dictionary-related data, as it can be rebuilt during decompression. As $n_{1}$ (dictionary size) and $n_{2}$ (input buffer size) directly influence the time complexity, memory requirements and compression ratio of the algorithm, as well as being recommended to be of value allowing to fully utilize the bits used to store them, these parameters require tuning, especially when used on a microcontroller. In order to store match position and match length efficiently, $n_{1}$ should be a power of two minus one, and $n_{2}$ should be a power of two. In this paper, match position and match length are stored as one byte each, with symbols stored as one byte by design. This directly translates into $n_{1}$ equal to 255 and $n_{2}$ equal to 256.

#### 3.3.3. Lempel–Ziv 78

Lempel–Ziv 78 (LZ78) [35] is a lossless data compression algorithm that is based on replacing the strings of symbols in data with tuples consisting of the index in a dictionary storing previous occurrences of strings of symbols and the next symbol from the input.

In terms of implementation, the last tuple can be denormalized to index only in case the next input symbol is the end of file (EOF)—storing EOF explicitly is not needed, as the dictionary index alone holds required information in this case. The dictionary index can be variable-width encoded, resulting in a better compression ratio—code used to calculate the minimal amount of bits required to uniquely represent a dictionary index with a given dictionary size is shown on the Listing 4. Symbols by design are stored as one byte each. The dictionary was implemented as a tree structure.

**Listing 4.** Function calculating the minimal amount of bits required to uniquely represent a dictionary index with a given dictionary size.
uint8_t bits(int dictionary_size)

{

    uint8_t i = dictionary_size;

    uint8_t result = 1;


    while (i >>= 1) result++;


    return result;

}


#### 3.3.4. Lempel–Ziv–Welch

Lempel–Ziv–Welch (LZW) [36] is a lossless data compression algorithm based on LZ78. In terms of implementation, the dictionary index is variable-width encoded in the same way as LZ78. The dictionary was implemented as a vector containing objects composed of a dictionary index and a string of symbols in the form of std::vector<uint8_t>. Due to the brute-force match search of this implementation, measured execution time and required energy were not acceptable. A second approach—utilizing a hash map—was tested, with the dictionary rewritten as std::unordered_map, with a string of symbols in the form of std::vector<uint8_t> as a key and dictionary index as value. Due to the fact that there is no implementation of std::hash<vector<uint8_t>> in C++, a custom hash solution for std::vector<uint8_t> was required—Fowler/Noll/Vo (FNV) hash algorithm [37] was chosen for this task, as utilizing std::hash<std::string> is not a resource-efficient approach.

The FNV algorithm [37] is a fast, lightweight hash algorithm that has a low collision rate. FNV-1a variant was chosen due to slightly better dispersion with smaller inputs. FNV-1a algorithm is shown on the Listing 5.

**Listing 5.** FNV-1a algorithm.
Initialize the hash variable, and name it H, with an offset basis.For each byte of data, name it B:–Set H to H XOR B–Multiply H by FNV_primeReturn H.


For the 32-bit hash version that was used, the offset basis is equal to 2166136261 (0x811C9DC5), and FNV_prime is equal to 16777619 (0x01000193). There are other FNV hash variants:FNV-1 is identical to FNV-1a, with differences in the order of XOR and multiplication operations.FNV-0 is identical to FNV-1, with differences in the hash initialization value of zero.

As stated earlier, FNV-1a hash was used instead of FNV-1 due to slightly better dispersion with smaller inputs. FNV-0 hash is deprecated, as it returns the same value for empty input and every input composed of any number of zero bytes. The offset basis for FNV-1 and FNV-1a algorithms was created by calculating the FNV-0 hash of the signature of one of the authors, shown on the Listing 6.

**Listing 6.** Signature of Landon Curt Noll, used to calculate offset basis for FNV-1 and FNV-1a.
chongo <Landon Curt Noll> /\../\


#### 3.3.5. Joint Photographic Experts Group

JPEG [30] is a very popular lossy data compression algorithm, intended to be used on images. Measurements were conducted using libjpeg implementation with a quality factor of 85.

### 3.4. Measurement Details

In order to measure the time and energy required for individual operations, a test bench was constructed. One of the microcontroller’s pins was used as output producing a signal similar to the gating signal in terms of concept, with high logical level during measured operation and low level otherwise. This signal was measured using an oscilloscope, along with the voltage drop across the 1 Ω shunt resistor to measure current using Ohm’s law. The current measuring method is shown in Figure 1. Only the required peripherals of the microcontroller were powered—that is—appropriate General-Purpose Input/Output (GPIO) ports in order to use the microcontroller’s pins, and UART for transmitting the size of compressed data that also served as an indicator if the compression was successful, as large enough data can exhaust the microcontroller’s memory with certain algorithms, causing the microcontroller to crash. The measurements were conducted using a maximum available clock speed equal to 100 MHz.

It should be noted that the measured current includes a 3.3 V voltage regulator quiescent current. This decision is caused by:The existence of ESP-01 current spikes of over 400 mA that—without mitigation—would cause the supply voltage to drop under 2.9 V if the measurements were conducted directly on a 3.3 V rail.FS1000A requires the supply of a voltage higher than 3.3 V with the Blackpill board requiring 3.3 V—requirement to measure two currents at the same time was simplified to measuring 5 V input current, with FS1000A powered from 5 V rail and everything else from 3.3 V rail.

Although the Blackpill board has a 3.3V low dropout (LDO) regulator (AP7343) that should produce similar results, it does not have sufficient current capabilities, as it is rated for a maximum of 300 mA at recommended conditions and 400 mA as the absolute maximum rating [38]. To solve this, a popular LM1117-3.3 breakout board was used. Due to the fact that the quiescent current of this chip is typically equal to 5 mA and can reach up to 10 mA [39], usage of this chip is not acceptable, as its quiescent current is of the same order of magnitude as Blackpill current consumption. In order to fix this issue, the MCP1825S33 chip was used, with a typical quiescent current of 120 μA and a maximum of 220 μA, and a guaranteed maximum current output of at least 500 mA [40]. Due to the fact that LM1117 and MCP1825 chips are not pin compatible [39,40], MCP1825S33 was soldered with its position shifted and one pin was connected using a soldering wick. As this power supply board also has a power status LED that is not needed for module operation, it was disconnected.

Due to the fact that FS1000A was powered with 5 V and for this reason required 5 V TTL logic levels in order to operate correctly, it was mandatory to use a level converter board in order to interface this module with the Blackpill board, as it operates at 3.3 V level logic.

Due to the existence of big current spikes that exceed 400 mA during the operation of ESP-01, two capacitors were added directly to the power pins of the board to mitigate the impedance of wires powering the module. The first capacitor serves as a high frequency, high current pulse buffer—a 100 nF Metalized Polypropylene (MPP) film capacitor was chosen. The second capacitor serves as a short-term energy buffer—a 100 μF aluminum electrolytic capacitor was used. This modification allowed a fault-free operation of the ESP-01 module.

## 4. Results

This section presents the performance evaluation of each tested algorithm in terms of attained results versus required time and energy, and each transmission module in terms of required time and energy in order to send data of a given size. In order to understand the observations better, three metrics were introduced:*Compression ratio* defined as the compressed data size divided by the original data size.*Time ratio* defined as the sum of time required for compressed data transmission and time required for data compression, divided by time required for original data transmission.*Energy ratio* is defined as the sum of energy required for compressed data transmission and energy required for data compression, divided by energy required for original data transmission.

It should be noted that a ratio of value less than 1.0 directly translates into a profit, meanwhile ratio of value greater than 1.0 directly translates into a loss. Compression ratio is used in conjunction with measured time and energy to evaluate compression algorithms. Time and energy ratios are used to compare the effectiveness of the “compress and send compressed data” scenario versus the “send uncompressed data” scenario, with different compression algorithms and transmission modules. Selected oscillograms from measurements related to data transmission are shown in Appendix A.

### 4.1. Data Transmission

This section presents the measurements of time and energy required to transmit data of various lengths with different transmission modules. The main goal of these measurements is to acquire data transmission results that are necessary for conducting calculations of time and energy ratios.

#### 4.1.1. FS1000A

Transmission measurements using the FS1000A module were conducted with 5 kbps speed, as 10 kbps transmission is affected by severe deterioration of signal integrity, which was discussed in Section 3.1. The byte value used for testing the FS1000A module, being the ‘A’ character, was chosen due to the fact that its binary representation—being 01000001—contains many consecutive zeros, resulting in testing the Manchester II encoded signal at its full bandwidth. Transmission of a single byte is shown in Figure A3a. Due to the fact that the current consumption of the test bench composed of the Blackpill board as the main processing unit and the FS1000A board as the transmission module is constant when averaged over the time of transmission of each input state, the following values have been used for energy calculations:37.457 mA for transmission of logical one.14.390 mA for transmission of logical zero.

Time and energy required for data transmission using test bench composed of the Blackpill board as main processing unit and the FS1000A board as transmission module, calculated using the Formulas (Equation 2) and (Equation 3), respectively, include the transmission of preamble and postamble of protocol shown on the Listing 1.
(2)t=3.5+D∗0.2[ms],
where D—data length in bytes.
(3)$$E=\frac{5*37.457*(2.5+D*0.1)+5*14.390*(1+D*0.1)}{1000}\;\left[\mathrm{mWs}\right],$$
where D—data length in bytes.

Using the Formulas (Equation 2) and (Equation 3) to calculate the time and energy required to transmit 10 B of data as an example, values of 5.5 ms and 0.799 mWs are calculated.

#### 4.1.2. nRF24L01+

Transmission measurements using the nRF24L01+ module were conducted with 250 kbps speed, as using lower speeds results in increased signal-to-noise ratio (SNR), allowing for successful communication over longer distances, which is often a desired factor in embedded systems. Due to the fact that the nRF24L01+ module supports a payload size of up to 32 B, transmission of each possible payload size was measured. Selected measurement oscillograms are shown in Figure A3b and Figure A4b. The time and energy required to transmit a payload of a given size are shown in Figure 2.

Transmission module reported the completion of transmission in payload size-agnostic time of 4.020 ms. The energy required for transmission of payload of a given size has a trend line that is positively correlated with payload size. This can be noticed on oscillograms, shown in Figure A3b and Figure A4b, as progressively longer time slices of increased current consumption with larger payload sizes.

#### 4.1.3. ESP-01

ESP-01 module is operating in 802.11 b/g/n standard, which means, it cannot instantly transmit data after being turned on. Four steps were identified as required in order to prepare the transmission module to be able to send data:Boot device.Connect to Access Point (AP).Obtain IP address with Dynamic Host Configuration Protocol (DHCP).Establish Transmission Control Protocol (TCP) connection.

Selected measurement oscillograms are shown in Figure A2. Measurement results of each step are shown in Table 4.

When using the ESP-01 board as a transmission module in a battery-powered embedded system, a special consideration of battery power capability is mandatory, as usage of this device can result in current spikes of over 400 mA—rendering some of the batteries unsuitable for this task. For example, a current spike of 412 mA was measured shortly after TCP communication was established. This can be observed in the oscillogram shown in Figure A2d.

Due to the fact that the ESP-01 module supports a payload size of up to 2048 B, the transmission of payload sizes of $2^{n}$ was measured, where 0≤n≤11. Selected measurement oscillograms are shown in Figure A4c–f. The time and energy required to transmit a payload of a given size are shown in Figure 3.

It can be noticed that payloads of sizes up to 1024 B are transmitted using one packet, and payload size of 2048 B is transmitted using two packets. This may be caused by a maximum transmission unit (MTU) of 1500 B for the Ethernet frame, which is specified in IEEE 802.3 standard [41].

As a result, the time and energy required for data transmission using a test bench composed of the Blackpill board as the main processing unit and the ESP-01 board as the transmission module included required preparation steps, and only 1024 B and 2048 B payload sizes were used.

#### 4.1.4. Summary

To sum up, a total of three transmission modules were tested: FS1000A, nRF24L01+ and ESP-01. The cheapest board—FS1000A—requires an external antenna and carefully selected data encoding in order to work properly. The most expensive board—ESP-01—requires a careful assessment of battery power capability, as usage of this device can result in current spikes of over 400 mA—rendering some of the batteries unsuitable. The best results in terms of energy required to transmit data were achieved by the nRF24L01+ board.

### 4.2. Weather Station Data Compression

This section presents the measurements of time and energy required to compress weather station data of various lengths with different compression algorithms. The main goal of these measurements is to acquire data compression results that are necessary for conducting calculations of time and energy ratios, as well as to gather compressed data sizes, required for calculating the compression ratios. In order to measure the algorithms’ performance with different data lengths, smaller data samples were produced by trimming the weather station data. This operation resulted in a test data set composed of 1, 2, 3, 4, 6, 8, 12 and 24 h of data.

As the Huffman algorithm is very lightweight in terms of memory requirements, it was successfully tested using a full weather station test data set. The Huffman algorithm resulted in a stable compression ratio between 0.529 and 0.542 on test data, requiring considerably low amounts of time and energy.

As the LZ77 algorithm is lightweight in terms of memory requirements due to the constant dictionary size, it was successfully tested using a full weather station test data set. The LZ77 algorithm resulted in a compression ratio ranging from 0.195 to 0.272, showing a decreasing trend with increasing input data size, at the cost of higher—albeit still in the low range—amounts of time and energy required when compared to the Huffman algorithm.

Due to the fact that the dictionary size of the LZ78 algorithm grows with data size, it was successfully tested using up to 6 h of weather station data, as larger data sizes resulted in an embedded system crash caused by not enough memory available. The LZ78 algorithm resulted in a compression ratio ranging from 0.274 to 0.389, showing a decreasing trend with increasing input data size. In terms of compression ratio, this algorithm produced results better than Huffman, and worse than LZ77. In terms of time and energy required for compression, this algorithm is more lightweight than LZ77, while being more demanding than Huffman.

Due to the fact that the dictionary size of the LZW algorithm grows with data size, it was successfully tested using up to 4 h of weather station data, as larger data sizes resulted in an embedded system crash caused by not enough memory available. The second approach trades a small amount of memory for vast speed improvement, as a hash map needs to store hash values in order to operate. This resulted in the maximum successfully tested data size lowered to 3 h of weather station data, as larger data sizes resulted in embedded system crashes caused by not enough memory available.

The LZW algorithm resulted in a compression ratio ranging from 0.229 to 0.313, showing a decreasing trend with increasing input data size. In terms of compression ratio, this algorithm produced results better than LZ78 and Huffman, and worse than LZ77. In terms of time and energy required for compression, first approach is not recommended due to time and energy requirements. Despite significant improvement in the second approach, it is still more demanding than Huffman, LZ77 and LZ78 in terms of time and energy requirements.

#### Comparison

The Huffman algorithm produced stable compression ratio results between 0.529 and 0.542 on test data. This can be explained due to the fact that this algorithm operates on symbol occurrence frequency, which does not change with input data length in a significant way. The LZ77 algorithm produced the best results of all tested algorithms in terms of compression ratio, as it leveraged the existence of repeating strings of symbols in data to compress it. The LZ78 algorithm produced slightly worse results than LZ77 in terms of compression ratio while outperforming Huffman by a significant margin. This was caused by the fact, that LZ78 excels at larger data sizes that are unattainable on the Blackpill board due to the available memory restrictions. The LZW algorithm also produced slightly worse results than LZ77 in terms of compression ratio, albeit with values closer to LZ77 than LZ78. This algorithm also excels at larger data sizes that are unattainable on the Blackpill board due to the available memory restrictions, hence the slightly worse results on relatively small input data sizes. The comparison of tested compression algorithms in terms of compression ratio is shown in Figure 4.

The computational complexity of tested algorithms can be observed by examining the time needed for compression, with the Huffman algorithm being the most lightweight, and LZW requiring the most time for compression for tested data sizes. The comparison of tested compression algorithms in terms of compression time is shown in Figure 5.

As the microcontroller consumes different amounts of current during the execution of various compression algorithms, it is also important to evaluate the energy required for each operation. The comparison of tested compression algorithms in terms of compression energy is shown in Figure 6.

### 4.3. Weather Station Data Transmission

This section presents the combined measurements of time and energy required for both scenarios—“send uncompressed data” and “compress and send compressed data”—using weather station data of various lengths with different compression algorithms and transmission modules. The main goal of these combined measurements is to calculate the saved time and energy by compressing and sending compressed data, compared to the scenario of uncompressed data transmission.

#### 4.3.1. FS1000A

In order to observe the impact of data compression on the energy consumption required for data transmission, transmission of original data was measured, as well as data compressed with tested algorithms. To assess the impact of data compression, time and energy ratios were evaluated. All algorithms except the first approach of LZW resulted in ratios below 1.0, directly translating into a profit. Usage of the LZ78 algorithm yielded the best time ratios across all data samples it was used on. In the case of energy ratios, LZ77 provided the best results for data samples up to 4 h in length, but with data samples containing 6 h of weather station data, LZ78 resulted in a slightly lower ratio, directly translating to energy savings. This means, in the case of an embedded system consisting of the Blackpill board as the main processing unit and the FS1000A board as the transmission module, in order to conserve energy it is most beneficial to use the LZ77 algorithm, except for the small edge of 0.002 for the LZ78 algorithm in ratio difference for data length of 6 h of weather station data. The comparisons of tested compression algorithms in terms of time and energy ratios are shown in Figure 7 and Figure 8, respectively.

To sum up, usage of the LZ78 algorithm resulted in the best time ratios across all tested data samples. In terms of energy ratios, LZ77 provided the best results for data samples up to 4 h in length, with LZ78 providing better results for 6 h data samples. Therefore, it is recommended to use these algorithms in the case of an embedded system comprised of the Blackpill board as the main processing unit and the FS1000A board as the transmission module in order to maximize the conservation of energy.

#### 4.3.2. nRF24L01+

In order to observe the impact of data compression on the energy consumption required for data transmission, the transmission of original data was measured, as well as data compressed with tested algorithms. To assess the impact of data compression, time and energy ratios were evaluated. All algorithms except the first approach of LZW, and the second approach of LZW with weather data length of 1 and 2 h, resulted in ratios below 1.0, directly translating into a profit. It should be noted that the second approach of LZW with weather data length of 1 and 2 h resulted in time ratios below 1.0. Usage of the LZ78 algorithm yielded the best time ratios across all data samples it was used on. In the case of energy ratios, LZ78 provided the best results for all data samples it was used on. This means, in the case of an embedded system consisting of the Blackpill board as the main processing unit and the nRF24L01+ board as the transmission module, in order to conserve energy it is most beneficial to use the LZ78 algorithm. The comparisons of tested compression algorithms in terms of time and energy ratios are shown in Figure 9 and Figure 10, respectively.

To sum up, usage of the LZ78 algorithm resulted in the best time and energy ratios across all tested data samples. Therefore, it is recommended to use this algorithm in the case of an embedded system comprised of the Blackpill board as the main processing unit and the nRF24L01+ board as a transmission module in order to maximize the conservation of energy.

#### 4.3.3. ESP-01

In order to observe the impact of data compression on the energy consumption required for data transmission, the transmission of original data was measured, as well as the data compressed with tested algorithms. To assess the impact of data compression, time and energy ratios were evaluated. All algorithms except the first approach of LZW resulted in ratios below 1.0, directly translating into a profit. Usage of the LZ77 algorithm yielded the best time ratios for data samples up to 2 h in length, but with data samples containing 3 or more hours of weather station data, LZ78 resulted in a slightly lower ratio, directly translating to time savings. In the case of energy ratios, LZ77 provided the best results for all data samples it was used on. This means, in the case of an embedded system consisting of the Blackpill board as the main processing unit and the ESP-01 board as the transmission module, in order to conserve energy it is most beneficial to use the LZ77 algorithm. The comparisons of tested compression algorithms in terms of time and energy ratios are shown in Figure 11 and Figure 12, respectively.

To sum up, the utilization of the LZ77 algorithm resulted in the best time ratios for data samples up to 2 h in length, with the LZ78 algorithm delivering better results for data samples containing 3 or more hours of data. In terms of energy ratios, LZ77 provided the best results across all tested data samples. Therefore, it is recommended to use this algorithm in the case of an embedded system comprised of the Blackpill board as the main processing unit and the ESP-01 board as the transmission module in order to maximize the conservation of energy.

#### 4.3.4. Summary

To conclude, in the case of the FS1000A transmission module, using the LZ78 algorithm resulted in the best time ratios across all tested data samples. In terms of energy ratios, LZ77 provided the best results for data samples up to 4 h in length, with LZ78 providing better results for 6 h data samples. Therefore, it is recommended to use these algorithms in the case of an embedded system comprised of the Blackpill board as the main processing unit and the FS1000A board as the transmission module in order to maximize the conservation of energy.

For the nRF24L01+ transmission module, using the LZ78 algorithm resulted in the best time and energy ratios across all tested data samples. Therefore, it is recommended to use this algorithm in the case of an embedded system comprised of the Blackpill board as the main processing unit and the nRF24L01+ board as the transmission module in order to maximize the conservation of energy.

Finally, regarding the ESP-01 transmission module, using the LZ77 algorithm yielded the best time ratios for data samples up to 2 h in length, while the LZ78 algorithm provided better results for data samples containing 3 or more hours of data. In terms of energy ratios, LZ77 provided the best results across all tested data samples. Therefore, it is recommended to use this algorithm in the case of an embedded system comprised of the Blackpill board as the main processing unit and the ESP-01 board as the transmission module in order to maximize the conservation of energy.

### 4.4. Image Data Compression

This section presents the measurements of time and energy required to compress image data of various image sizes. The main goal of these measurements is to acquire image data compression results that are necessary for conducting calculations of time and energy ratios, as well as to gather compressed image data sizes, required for calculating the compression ratios. In order to measure the algorithm’s performance with different image sizes, three sizes commonly found in embedded devices were chosen: 160 × 120 px, 320 × 240 px and 640 × 480 px.

#### JPEG

The JPEG algorithm was successfully tested with all image sizes. Measurement results are shown in Table 5. In order to further evaluate the performance of the JPEG algorithm, a validation subset of the COCO dataset [42] (2017 updated version) was used, consisting of 5000 photos of various classes. Each photo was resized to all sizes evaluated in this paper (160 × 120 px, 320 × 240 px and 640 × 480 px) and compressed using the same quality factor of 85.

To sum up, the JPEG algorithm resulted in a compression ratio ranging from 0.028 to 0.037—an order of magnitude smaller than Huffman, LZ77, LZ78 and LZW—while being considerably lightweight. Even in the worst-case scenario from the COCO dataset, the maximum observed compression ratio is equal to 0.212, which is an impressive result.

### 4.5. Image Data Transmission

This section presents the combined measurements of time and energy required for both scenarios—“send uncompressed data” and “compress and send compressed data”—using image data of various image sizes with different transmission modules. The main goal of these combined measurements is to calculate the saved time and energy by compressing and sending compressed data, compared to the scenario of uncompressed data transmission.

#### 4.5.1. FS1000A

In order to observe the impact of data compression on the energy consumption required for data transmission, the transmission of original data was measured, as well as data compressed with the tested algorithm. To assess the impact of data compression, time and energy ratios were evaluated. The JPEG algorithm resulted in ratios vastly below 1.0, directly translating into a profit. This means, in the case of an embedded system consisting of the Blackpill board as the main processing unit and the FS1000A board as the transmission module, in order to conserve energy it is crucial to use the JPEG algorithm to compress image data. Visual representations of time and energy ratios are shown in Figure 13 and Figure 14.

In summary, the JPEG algorithm produced energy ratios ranging from 0.020 to 0.033. Even in the worst-case scenario from the COCO dataset, the maximum observed energy ratio is equal to 0.220, which is an impressive result. Therefore, it is definitely recommended to use this algorithm in the case of an embedded system comprised of the Blackpill board as the main processing unit and the FS1000A board as a transmission module in order to maximize the conservation of energy.

#### 4.5.2. nRF24L01+

In order to observe the impact of data compression on the energy consumption required for data transmission, the transmission of original data was measured, as well as data compressed with the tested algorithm. To assess the impact of data compression, time and energy ratios were evaluated. The JPEG algorithm resulted in ratios vastly below 1.0, directly translating into a profit. This means, in the case of an embedded system consisting of the Blackpill board as the main processing unit and the nRF24L01+ board as the transmission module, in order to conserve energy it is crucial to use the JPEG algorithm to compress image data. Visual representations of time and energy ratios are shown in Figure 15 and Figure 16.

In summary, the JPEG algorithm produced energy ratios ranging from 0.040 to 0.049. Even in the worst-case scenario from the COCO dataset, the maximum observed energy ratio is equal to 0.226, which is an impressive result. Therefore, it is definitely recommended to use this algorithm in the case of an embedded system comprised of the Blackpill board as the main processing unit and the nRF24L01+ board as the transmission module in order to maximize the conservation of energy.

#### 4.5.3. ESP-01

In order to observe the impact of data compression on the energy consumption required for data transmission, the transmission of original data was measured, as well as data compressed with the tested algorithm. To assess the impact of data compression, time and energy ratios were evaluated. The JPEG algorithm resulted in ratios vastly below 1.0, directly translating into a profit. This means, in the case of an embedded system consisting of the Blackpill board as the main processing unit and the ESP-01 board as the transmission module, in order to conserve energy it is crucial to use the JPEG algorithm to compress image data. Visual representations of time and energy ratios are shown in Figure 17 and Figure 18.

To sum up, the JPEG algorithm resulted in an energy ratio ranging from 0.040 to 0.174. Even in the worst-case scenario from the COCO dataset, the maximum observed energy ratio is equal to 0.311, which is an impressive result. Therefore, it is definitely recommended to use this algorithm in the case of an embedded system comprised of the Blackpill board as the main processing unit and the ESP-01 board as the transmission module in order to maximize the conservation of energy.

#### 4.5.4. Summary

In summary, for the FS1000A transmission module, the JPEG algorithm yielded energy ratios ranging from 0.020 to 0.033. Even in the worst-case scenario from the COCO dataset, the maximum observed energy ratio is equal to 0.220, which is an impressive result. Therefore, it is definitely recommended to use this algorithm in the case of an embedded system comprised of the Blackpill board as the main processing unit and the FS1000A board as the transmission module in order to maximize the conservation of energy.

Regarding the nRF24L01+ transmission module, the JPEG algorithm resulted in an energy ratio ranging from 0.040 to 0.049. Even in the worst-case scenario from the COCO dataset, the maximum observed energy ratio is equal to 0.226, which is an impressive result. Therefore, it is definitely recommended to use this algorithm in the case of an embedded system comprised of the Blackpill board as the main processing unit and the nRF24L01+ board as a transmission module in order to maximize the conservation of energy.

For the ESP-01 transmission module, the JPEG algorithm resulted in an energy ratio ranging from 0.040 to 0.174. Even in the worst-case scenario from the COCO dataset, the maximum observed energy ratio is equal to 0.311, which is an impressive result. Therefore, it is definitely recommended to use this algorithm in the case of an embedded system comprised of the Blackpill board as the main processing unit and the ESP-01 board as the transmission module in order to maximize the conservation of energy.

## 5. Discussion

According to the measurements and results shown in this article, it is possible to lower the energy consumption required for data transmission in a microcontroller-based system by compressing the data using carefully selected algorithms. This results in a prolonged lifespan of battery-powered TinyML devices, minimizing maintenance costs due to less frequent battery replacements. For example, with an energy ratio of 0.2, five times more data can be transmitted, resulting in a five times longer battery lifespan. It is also an important factor in the case of systems located in hard-to-reach locations—for example, inside an industrial machine—as less frequent battery replacements directly translate into lower total machine downtime.

It should be noted, that considering the existing TinyML studies and devices, to the best of our knowledge, there is no TinyML research paper that evaluates the impact of data compression on the energy consumption required for data transmission in a microcontroller-based system with resources comparable to the used STM32F411CE chip. The main conclusion of this paper is that by carefully selecting a compression algorithm and using it to compress the data before transmission, a significant amount of energy can be saved. By using the best approach for weather station data, that is, the nRF24L01+ board and the LZ78 algorithm, a 0.452 energy ratio can be achieved with 6 h of data, translating into a 2.21 times longer battery lifespan. For the image data, by using the ESP-01 board and the JPEG algorithm, the energy ratio ranged from 0.040 to 0.174, translating into a 5.74–25 times longer battery lifespan.

In the case of the FS1000A transmission module and weather data, using the LZ78 algorithm resulted in the best time ratios across all tested data samples. In terms of energy ratios, LZ77 provided the best results for data samples up to 4 h in length, with LZ78 providing better results for 6 h data samples. Therefore, it is recommended to use these algorithms in the case of an embedded system comprised of the Blackpill board as the main processing unit and the FS1000A board as the transmission module in order to maximize the conservation of energy.

Regarding the nRF24L01+ transmission module and weather data, usage of the LZ78 algorithm resulted in the best time and energy ratios across all tested data samples. Therefore, it is recommended to use this algorithm in the case of an embedded system comprised of the Blackpill board as the main processing unit and the nRF24L01+ board as the transmission module in order to maximize the conservation of energy.

For the ESP-01 transmission module and weather data, using the LZ77 algorithm resulted in the best time ratios for data samples up to 2 h in length, with the LZ78 algorithm delivering better results for data samples containing 3 or more hours of data. In terms of energy ratios, LZ77 provided the best results across all tested data samples. Therefore, it is recommended to use this algorithm in the case of an embedded system comprised of the Blackpill board as the main processing unit and the ESP-01 board as the transmission module in order to maximize the conservation of energy.

Regarding the FS1000A transmission module and image data, the JPEG algorithm resulted in an energy ratio ranging from 0.020 to 0.033. Even in the worst-case scenario from the COCO dataset, the maximum observed energy ratio is equal to 0.220, which is an impressive result. Therefore, it is definitely recommended to use this algorithm in the case of an embedded system comprised of the Blackpill board as the main processing unit and the FS1000A board as a transmission module in order to maximize the conservation of energy.

In the case of the nRF24L01+ transmission module and image data, the JPEG algorithm resulted in an energy ratio ranging from 0.040 to 0.049. Even in the worst-case scenario from the COCO dataset, the maximum observed energy ratio is equal to 0.226, which is an impressive result. Therefore, it is definitely recommended to use this algorithm in the case of an embedded system comprised of the Blackpill board as the main processing unit and the nRF24L01+ board as a transmission module in order to maximize the conservation of energy.

Moreover, regarding the ESP-01 transmission module and image data, the JPEG algorithm resulted in an energy ratio ranging from 0.040 to 0.174. Even in the worst-case scenario from the COCO dataset, the maximum observed energy ratio is equal to 0.311, which is an impressive result. Therefore, it is definitely recommended to use this algorithm in the case of an embedded system comprised of the Blackpill board as the main processing unit and the ESP-01 board as the transmission module in order to maximize the conservation of energy.

To summarize, the nRF24L01+ module is suggested for a battery-powered embedded device that will transmit data, since it needs the least amount of energy to transmit one byte of data. For weather data, the nRF24L01+ module is best combined with the LZ78 algorithm in order to achieve the best results, requiring only 29.061 mWs in order to compress and transmit 1 h of weather station data, and 137.623 mWs for 6 h of weather station data. In order to calculate the theoretical maximum amount of transmissions using one standard 18,650 battery with a nominal voltage of 3.7 V and capacity of 2500 mAh, energy amounts can be recalculated to 3.7 V supply voltage, resulting in 21.505 mWs for 1 h of weather station data and 101.841 mWs for 6 h of weather station data. As the mentioned 18,650 battery contains 9250 mWh of energy, this results in 1,548,467 possible transmissions containing 1 h of weather station data and 326,980 possible transmissions containing 6 h of weather station data, translating into 1,548,467 h and 1,961,880 h worth of transmitted data, respectively. Due to the fact that this is equal to over 176 years and over 223 years respectfully, the amount of consumed energy over time is smaller than the self-discharge of such a battery, leaving plenty of energy for other purposes including collecting data from multiple sensors, processing data, and inferring TinyML models.

## 6. Conclusions

This paper presents an analysis of the impact of data compression on the energy consumption required for data transmission in a system based on a battery-powered microcontroller. The resource-constrained system under study is equipped with an STM32F411CE chip and a transmission module. We have demonstrated that by carefully selecting an algorithm for compressing various types of data before transmission, a significant amount of energy can be conserved. To create a battery-powered embedded device for data transmission, we recommend the use of the nRF24L01+ module, as it requires the least amount of energy for transmission. For specific data types like weather data, the nRF24L01+ module exhibits the best performance when paired with the LZ78 compression algorithm, providing 0.452 with 6 h of data, translating into a 2.21 times longer battery lifespan. In turn, for the image data, the application of the ESP-01 board and the JPEG algorithm provides an energy ratio in the range from 0.040 to 0.174, which yields a 5.74–25 times longer battery lifespan.

It is essential to emphasize that the energy saved through data compression can be redirected to other operations, including collecting data from multiple sensors, processing data, and notably, inferring TinyML models. While the implementation of artificial intelligence methods reduces operating time, it introduces new functional possibilities for the device.

There is still potential for further improvements. Future work may involve measurements of different compression algorithms for text and image data. Additionally, research could include further measurements of the JPEG algorithm, focusing on tuning the quality factor to maximize the benefits of data compression while maintaining an acceptable level of image quality, dictated by the specific use case. Further studies could also explore other hardware options, encompassing both microcontrollers and transmission modules, that can meet the stringent limitations of TinyML devices. Finally, a very interesting direction for future research is cybersecurity, as it poses a serious threat to microcontroller-based systems analyzed in our manuscript. Moreover, exploring how cryptographic algorithms influence this consumption is another interesting topic for future work.

## Figures and Tables

**Figure 1 sensors-24-00224-f001:**
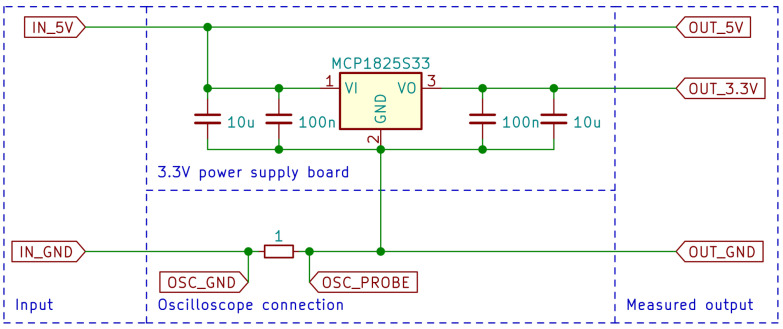
Measuring current method.

**Figure 2 sensors-24-00224-f002:**
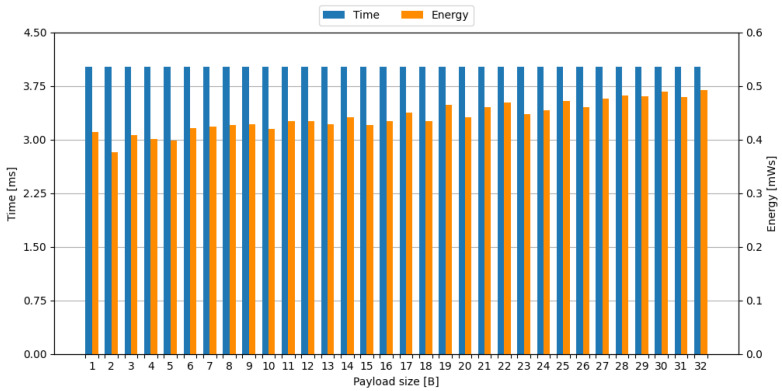
Time and energy required to transmit a payload of given size with nRF24L01+ transmission module.

**Figure 3 sensors-24-00224-f003:**
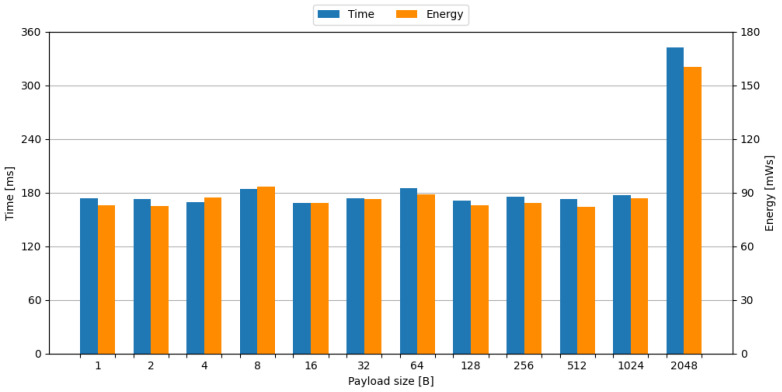
Time and energy required to transmit a payload of given size with the ESP-01 transmission module.

**Figure 4 sensors-24-00224-f004:**
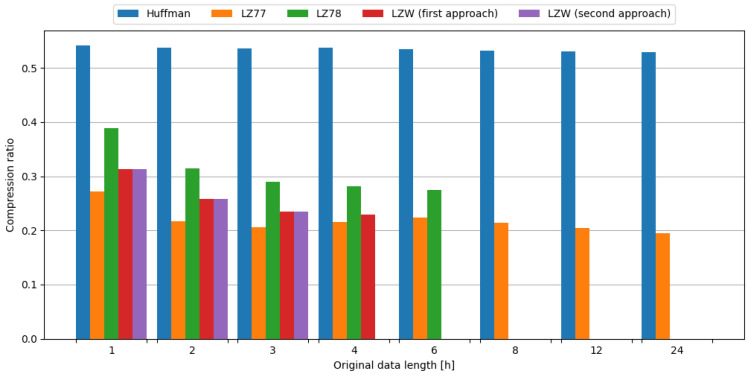
Comparison of compression algorithms tested on weather station data in terms of compression ratio.

**Figure 5 sensors-24-00224-f005:**
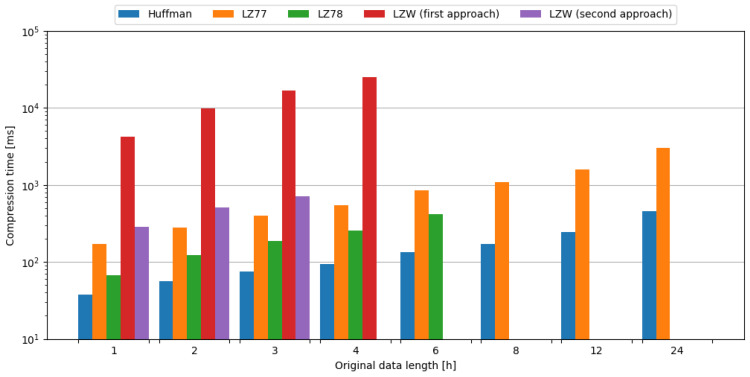
Comparison of compression algorithms tested on weather station data in terms of compression time.

**Figure 6 sensors-24-00224-f006:**
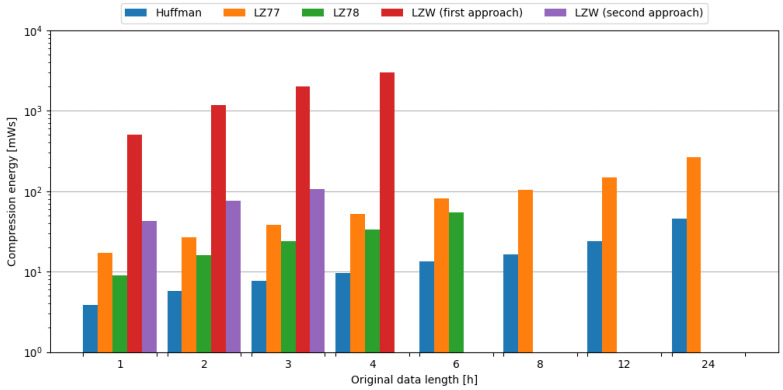
Comparison of compression algorithms tested on weather station data in terms of compression energy.

**Figure 7 sensors-24-00224-f007:**
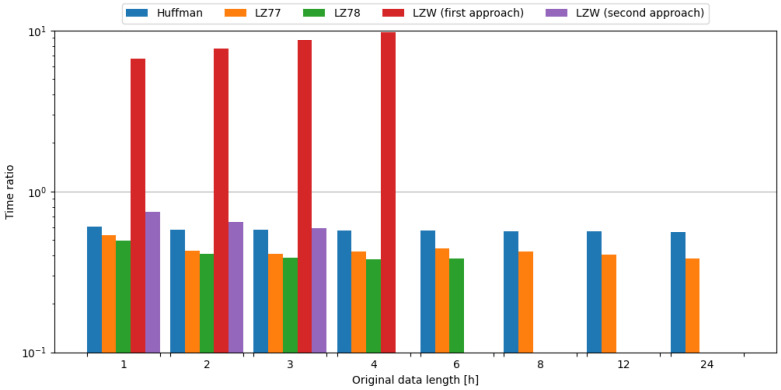
Comparison of compression algorithms tested on weather station data and the FS1000A board in terms of time ratio.

**Figure 8 sensors-24-00224-f008:**
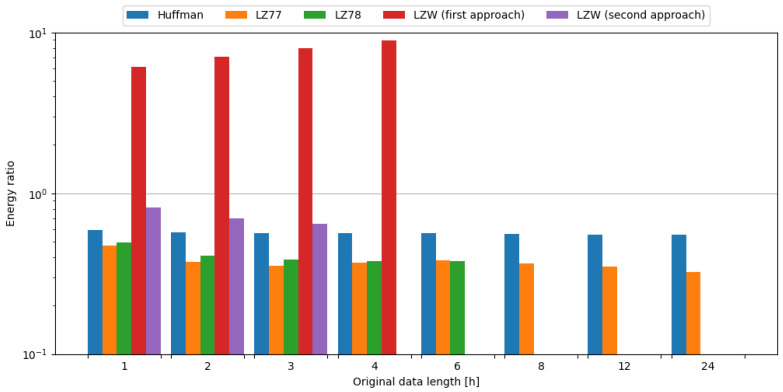
Comparison of compression algorithms tested on weather station data and the FS1000A board in terms of energy ratio.

**Figure 9 sensors-24-00224-f009:**
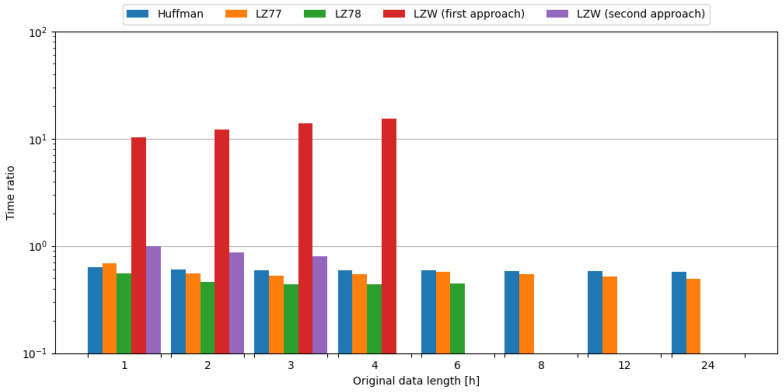
Comparison of compression algorithms tested on weather station data and the nRF24L01+ board in terms of time ratio.

**Figure 10 sensors-24-00224-f010:**
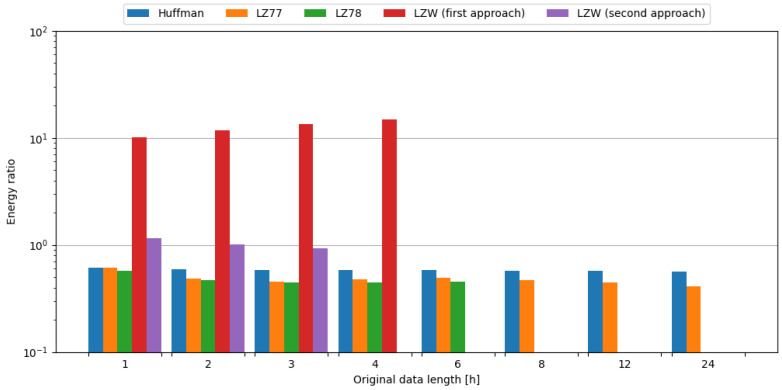
Comparison of compression algorithms tested on weather station data and the nRF24L01+ board in terms of energy ratio.

**Figure 11 sensors-24-00224-f011:**
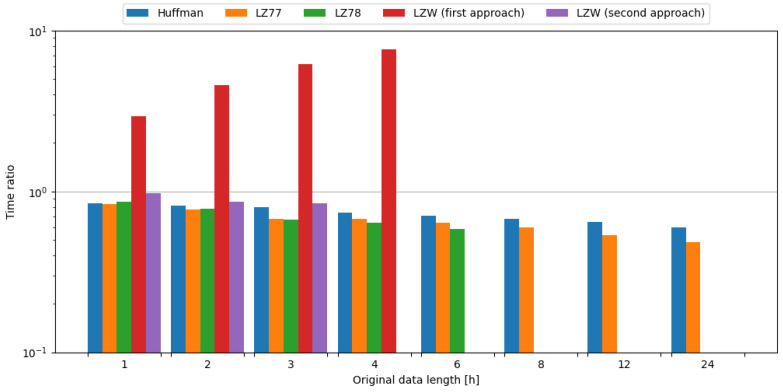
Comparison of compression algorithms tested on weather station data and the ESP-01 board in terms of time ratio.

**Figure 12 sensors-24-00224-f012:**
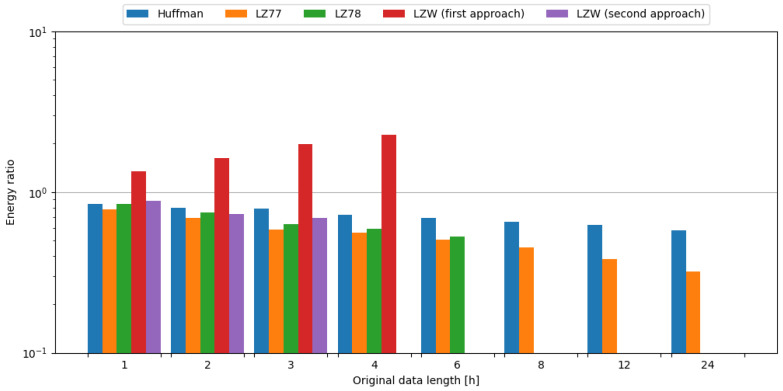
Comparison of compression algorithms tested on weather station data and the ESP-01 board in terms of energy ratio.

**Figure 13 sensors-24-00224-f013:**
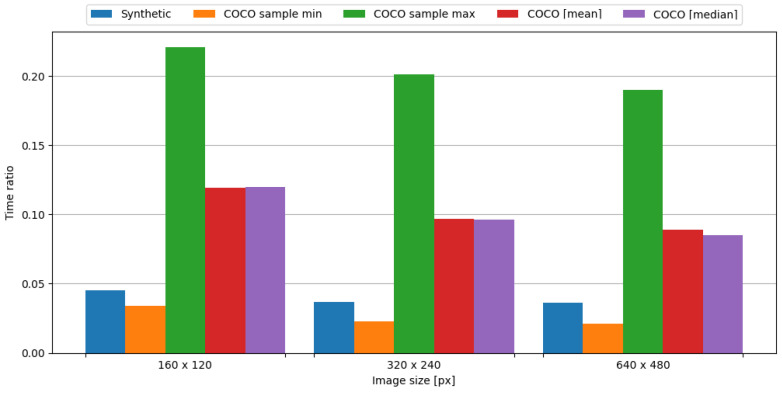
Time ratio with image data and the FS1000A board.

**Figure 14 sensors-24-00224-f014:**
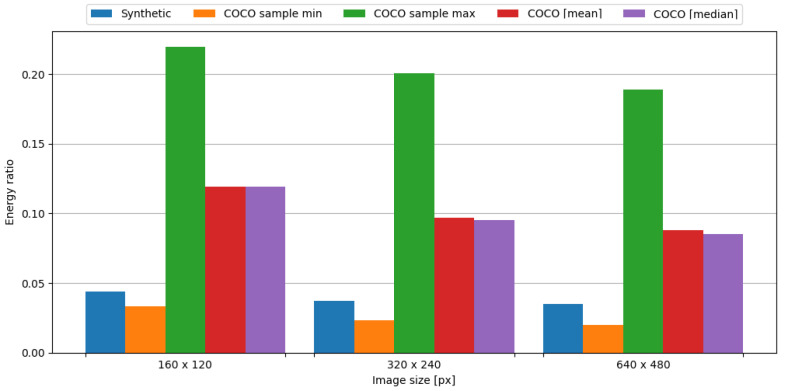
Energy ratio with image data and the FS1000A board.

**Figure 15 sensors-24-00224-f015:**
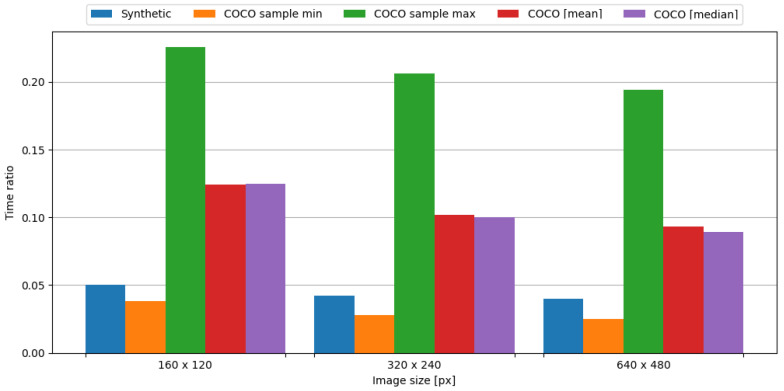
Time ratio with image data and the nRF24L01+ board.

**Figure 16 sensors-24-00224-f016:**
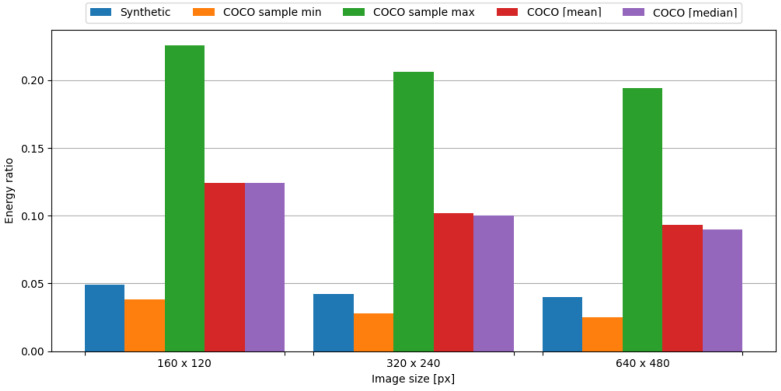
Energy ratio with image data and the nRF24L01+ board.

**Figure 17 sensors-24-00224-f017:**
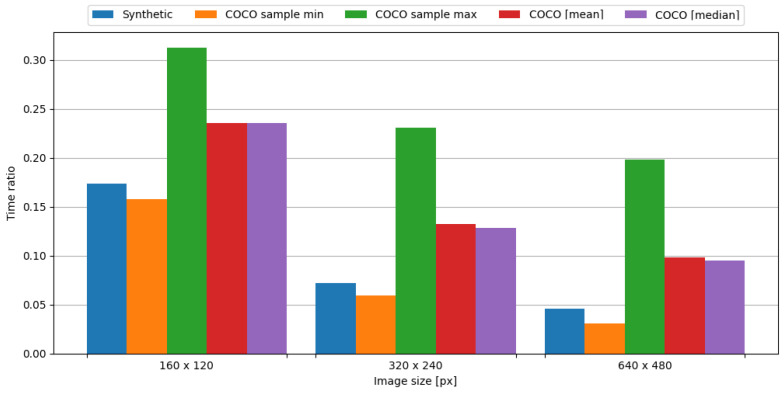
Time ratio with image data and ESP-01 board.

**Figure 18 sensors-24-00224-f018:**
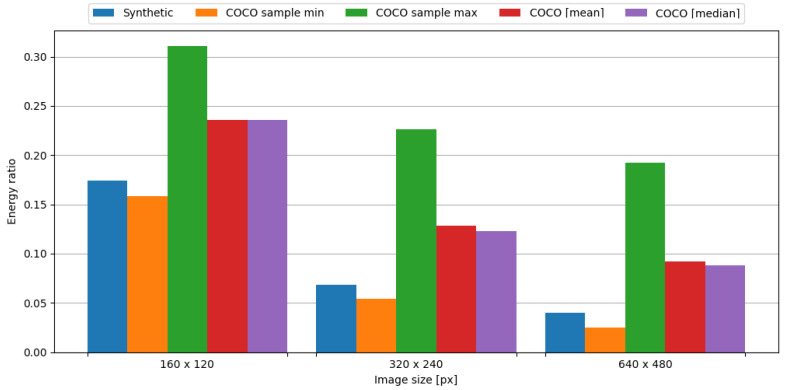
Energy ratio with image data and ESP-01 board.

**Table 1 sensors-24-00224-t001:** Transmission modules’ details.

	FS1000A	nRF24L01+	ESP-01
Supply voltage	3.5–12 V	1.9–3.6 V	3.3 V
Interface	TTL	SPI	UART with AT commands
Transmission speed	up to 10 kbps	up to 2 Mbps	802.11 b/g/n
Frequency band	433 MHz	2.4 GHz ISM	2.4 GHz ISM
Signal modulation	ASK	GFSK	802.11 b/g/n

**Table 2 sensors-24-00224-t002:** BME280 measurement ranges.

Parameter	Range
Temperature	−40…+85 ${}^{\circ }$C
Pressure	300…1100 hPa
Relative humidity	0…100 %

**Table 3 sensors-24-00224-t003:** Sections of the weather data format.

Section	Description	Example
YYYY-MM-DD	Current date	2023-03-22
HH:mm	Current time	13:25
NNNNNNNN	8-byte sensor name	Sensor 2
T: tt.tt	Temperature	T: 21.53
P: pppp.pp	Pressure	P: 995.95
H: hh.hh	Relative humidity	H: 66.57
\r\n	CR LF line break	

**Table 4 sensors-24-00224-t004:** Time and energy are required for each preparation step of the ESP-01 transmission module.

Preparation Step	Operation Time	Operation Energy
Boot device	185.650 ms	75.805 mWs
Connect to AP	155.315 ms	85.343 mWs
Obtain IP address with DHCP	788.140 ms	386.147 mWs
Establish TCP connection	189.404 ms	106.295 mWs

**Table 5 sensors-24-00224-t005:** Image data compression—JPEG algorithm.

Image Size	Original Data Size	Compressed Data Size	Compression Ratio	Compression Time	Compression Energy
160 × 120 px	57,600 B	2105 B	0.037	93.526 ms	11.353 mWs
320 × 240 px	230,400 B	6783 B	0.029	354.482 ms	43.406 mWs
640 × 480 px	921,600 B	25,941 B	0.028	1408.710 ms	173.072 mWs

## Data Availability

The data are available at: https://github.com/tobiaszpuslecki/Data_ImpactOfCompression/ (accessed on 19 October 2023).

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
