# Peer review of "Study of the Impact of Data Compression on the Energy Consumption Required for Data Transmission in a Microcontroller-Based System"

_sensors, 2023, doi:10.3390/s24010224_

Round 1
Reviewer 1 Report
Comments and Suggestions for Authors
In the manuscript, the authors have studied the impact of data compression on the energy consumption required for data transmission. Specifically, STM32F411CE is adopted as the microprocessor. Three transceiver modules are considered, including FS1000A, nRF24L01+, and ESP-01. For normal sensing data, four compression algorithms are involved, including Huffman, Lempel-Ziv 77, Lempel-Ziv 78, and Lempel-Ziv-Welch. While for image data, JPEG is used. Experiments have been conducted for the measurements of time and energy consumption. However, some concerns still need to be addressed.
1. The manuscript compares three different transceiver modules. However, the comparison is not quite fair. The modules work in different communications protocols with different working frequencies and communication ranges, which are suitable for different scenarios. They are not a competitive relationship.
2. In terms of the compression analysis, is it possible that a high compression ratio leads to a high data restoration error?
3. From Line 137 to Line 139, “Therefore, to fill this research gap, in the following sections of this paper we present a study of the impact of data compression on the energy consumption required for data transmission in a microcontroller-based.” The sentence seems to be incomplete.
4. From Line 143 to Line 147, “As there are various compression algorithms, each with its own benefits and drawbacks, a selection of simple, well-known algorithms was chosen because of relatively low amount of resources available on chosen embedded platform, as well as the presence of important requirement of minimizing the time and energy consumption which can be satisfied in an easier way by utilizing lightweight methods.” The sentence is not readable enough.
5. Instead of a research paper, the manuscript is more like a lab report. Could the authors please reorganize this work to highlight the technical contributions of this work and also make the summaries easier to follow?
6. Instead of the synthetic weather data and images, would it be better to analyze the microcontroller and transceiver modules with a practical scenario? Due to IoT is application-oriented, the selection of modules and algorithms is highly related to the specific applications.
Comments on the Quality of English LanguageSerious proofreading is needed. There are some typos and grammar errors.
Author Response
We sincerely appreciate the anonymous reviewer for his/her helpful and valuable comments for enriching and improving the quality and importance of the manuscript. The entire manuscript has been carefully revised according to reviewer’s comments. The most important changes to the revised manuscript are highlighted with a red color. Point-to-point responses to the specific questions and queries are presented below. Moreover, according to the remarks of reviewers, the manuscript was shortened, in more detail:
- Section 3.1 was synthesized, i.e., some descriptions were condensed or removed.
- Section 3.3 was synthesized, i.e., some descriptions were condensed or removed.
- Table 4 of the original manuscript was converted to a figure (Figure 2 in the revised manuscript).
- Table 6 of the original manuscript was converted to a figure (Figure 3 in the revised manuscript).
- Listing 7 and description of this listing was removed.
- Section 5 was synthesized, i.e., some descriptions were condensed or removed.
- The manuscript compares three different transceiver modules. However, the comparison is not quite fair. The modules work in different communications protocols with different working frequencies and communication ranges, which are suitable for different scenarios. They are not a competitive relationship.
Response:
We thank the reviewer for this comment. Yes, the modules differ vastly in terms of parameters such as operating frequency, protocols and communication ranges. The comparison of these modules is not the main takeway of the paper – the goal was to check the effect of compression algorithms on data transmission in a microcontroller system. Evaluation of different transmission modules can prove that the positive effect of the compression is present regardless of the used transmission module.
- In terms of the compression analysis, is it possible that a high compression ratio leads to a high data restoration error?
Response:
We thank the reviewer for this comment. Yes, data compressed with high compression ratio can theoretically lead to higher than usual data restoration errors in case of significant data transmission errors occurrence. However, usage of data correction codes (except those already present in modules) is outside the scope of our paper.
- From Line 137 to Line 139, “Therefore, to fill this research gap, in the following sections of this paper we present a study of the impact of data compression on the energy consumption required for data transmission in a microcontroller-based.” The sentence seems to be incomplete.
Response:
We thank the reviewer for this comment. We corrected the sentence as follows:
“Therefore, to fill this research gap, in the following sections of this paper, we present a study of the impact of data compression on the energy consumption required for data transmission in a microcontroller-based system.”
- From Line 143 to Line 147, “As there are various compression algorithms, each with its own benefits and drawbacks, a selection of simple, well-known algorithms was chosen because of relatively low amount of resources available on chosen embedded platform, as well as the presence of important requirement of minimizing the time and energy consumption which can be satisfied in an easier way by utilizing lightweight methods.” The sentence is not readable enough.
Response:
We thank the reviewer for this comment. We corrected the sentence as follows:
“There are various compression algorithms, each with its own benefits and drawbacks. A selection of simple, well-known algorithms was chosen because of relatively low amount of resources available on chosen embedded platform. The presence of important requirement of minimizing the time and energy consumption can be satisfied in an easier way by utilizing lightweight methods.”
- Instead of a research paper, the manuscript is more like a lab report. Could the authors please reorganize this work to highlight the technical contributions of this work and also make the summaries easier to follow?
Response:
We thank the reviewer for this comment. We reformulated some elements the abstract, introduction and conclusions to highlight the technical contributions of this work and also make the summaries easier to follow. Moreover, we have shortened Section 5. “Discussion” and we have left only the most important takeaways.
- Instead of the synthetic weather data and images, would it be better to analyze the microcontroller and transceiver modules with a practical scenario? Due to IoT is application-oriented, the selection of modules and algorithms is highly related to the specific applications.
Response:
We thank the reviewer for this comment. When we started our work on the manuscript, after long discussions and analysis we decided to make our research for two types of uses cases that in our opinion are quite representative for IoT applications: weather data and images. Due to the paper size, we have not added additional uses cases.
Weather data was not synthetic, i.e., real data was used. We selected this use case as an example of a system that transmits simple data such as numbers or text.
In turn, in the second use case, the tested images were synthetic due to the constraint of memory size of the microcontroller, making it not possible to store raw RGB888 640x480 px image. Measuring the scenario with real camera would also include the time and current required for the camera operation, image data acquisition and compression. A synthetic image solves this issue allowing for the the measurement of compression exclusively. The computational complexity of JPEG algorithm is O(n), where n is the number of pixels in the image. This means, time and current measurements should not be significantly different. The compression ratio, however, will differ - COCO dataset was used to check the best, worst, mean and median cases of possible real image data.
Comments on the Quality of English Language
Serious proofreading is needed. There are some typos and grammar errors.
Response:
We thank the reviewer for this comment. We did a detailed proofreading of the manuscript to correct typos and grammar errors.
Reviewer 2 Report
Comments and Suggestions for Authors
The authors present a detailed analysis of meaningful metrics to evaluate the time and energy costs of transmitting typical data in battery-powered IoT devices. After the investigation and the evaluation of the hardware resources (ARM Cortex-M processor and transmission modules), they perform a comparison of the highlighted performance between the transmission of data without compression and the transmission of data when applying some compression algorithms to reduce the payload, including in the metrics related to the time and energy costs also the corresponding costs due to the compression algorithms. This approach gives an elevated scientific soundness to the work that results being very clear and well-organized. Indeed, the English grammar and spelling are proper, the equations are informative and easy to understand, as well as figures and tables that support the text and the comprehension of the work. The cited publications are relevant to the work, and the authors describe the relevance of their work in the corresponding research field. In addition, because of the applied approach, the methods and the material used are logically described and the results are convincing.
Some minors need to be addressed to improve the quality of the work: for this purpose refer to the attached file.

Author Response
We sincerely appreciate the anonymous reviewer for his/her helpful and valuable comments for enriching and improving the quality and importance of the manuscript. The entire manuscript has been carefully revised according to reviewer’s comments. The most important changes to the revised manuscript are highlighted with a red color. Point-to-point responses to the specific questions and queries are presented below. Moreover, according to the remarks of reviewers, the manuscript was shortened, in more detail:
- Section 3.1 was synthesized, i.e., some descriptions were condensed or removed.
- Section 3.3 was synthesized, i.e., some descriptions were condensed or removed.
- Table 4 of the original manuscript was converted to a figure (Figure 2 in the revised manuscript).
- Table 6 of the original manuscript was converted to a figure (Figure 3 in the revised manuscript).
- Listing 7 and description of this listing was removed.
- Section 5 was synthesized, i.e., some descriptions were condensed or removed.
Some minors need to be addressed to improve the quality of the work:
As well as figure(s) and table(s), also the sections should be referenced using the word with the capital initial in the text, i.e. 'Section'. In addition, they should be all referenced using the numbering: this is not done for Section 6 ('Conclusions') at line 79, whereas Section 5 ('Discussion') is never referenced.
Response:
We thank the reviewer for this comment. We corrected the paper according to the provided remarks.
Since cybersecurity is a serious threat to all modern digital applications and digital systems, this aspect should be included because security protection mechanisms are fundamental also in sensor-based loT applications. The application of security protections affects both time and energy costs as well as the compression algorithms used by the authors in this work. My suggestion is not to include also such costs in the presented investigation at this stage, anyway this aspect would highly improve the quality of the work, hence, at this stage, my suggestion is to at least mention this aspect by including it in the Section 5 that is dedicated to the 'discussion' and citing relevant works. For instance, the work [1] reported below presents a comparison between hardware and software implementations of the main typical security protection mechanisms in the same terms of time and energy consumption and using an approach similar to the one used in this work. In particular [1] presents also the costs in terms of time and energy consumption for the software implementation of security functions that are executed on an ARM Cortex processor.
[1] Baldanzi, Luca, et al. "Crypto accelerators for power-efficient and real-time on-chip implementation of secure algorithms." 2019 26th IEEE International Conference on Electronics, Circuits and Systems (ICECS). IEEE, 2019.
Response:
We thank the reviewer for this comment. We fully agree with the reviewer that cybersecurity is a serious threat to microcontroller-based system analyzed in our manuscript. We added in Conclusions this topic as a potential future direction in our research. However, since other reviewers pointed out that the manuscript is too long, we have not added a longer discussion on this topic.
Reviewer 3 Report
Comments and Suggestions for Authors
The main objective of this work is to study the impact of data compression on the power consumption of data compression on the power consumption of data transmission, using different data transmission, using different transmission modules, in a microcontroller-based system with very limited resources. The authors study the performance of different compression algorithms and transmission modules, taking into account the computational and memory complexity, as well as a performance analysis.
The proposal is to use an embedded device that transmits sensor data, based on the STM32F411 microcontroller. The STM32F411CE microcontroller is to be used as the transmitter module, and the nRF24L01+ board is to be used as the transmitter module, since it requires the least amount of memory. module, since it requires the least amount of power to transmit one byte of data. The authors conclude that the best option is to use the LZ78 algorithm in combination to achieve the highest energy and time efficiency.
In the case of image data, using the JPEG algorithm for compression gives the best results.
The article is too long and I think there are many things that can be synthesised.
Things that can be synthesised. I would include a description of the material including a table and for more detail I would either add an appendix or upload it to github with the I would upload it to github along with the full description of the experiment.
Also, I think there are too many numerical tables that could be condensed into more representative figures that are already included in the paper.
The work is very interesting and represents a significant and novel work, although I would recommend at least a look at the previous comments regarding the structure.
On the other hand, I would recommend at least exploring how cryptographic algorithms influence cryptographic algorithms influence this consumption. Another limitation of IoT devices related to their reduced capabilities is security, as shown in "A Test Environment for Wireless Hacking in Domestic IoT Scenarios". IoT Scenarios'. I suggested exploring this as a potential future work.
Comments on the Quality of English Language
The main objective of this work is to study the impact of data compression on the power consumption of data compression on the power consumption of data transmission, using different data transmission, using different transmission modules, in a microcontroller-based system with very limited resources. The authors study the performance of different compression algorithms and transmission modules, taking into account the computational and memory complexity, as well as a performance analysis.
The proposal is to use an embedded device that transmits sensor data, based on the STM32F411 microcontroller. The STM32F411CE microcontroller is to be used as the transmitter module, and the nRF24L01+ board is to be used as the transmitter module, since it requires the least amount of memory. module, since it requires the least amount of power to transmit one byte of data. The authors conclude that the best option is to use the LZ78 algorithm in combination to achieve the highest energy and time efficiency.
In the case of image data, using the JPEG algorithm for compression gives the best results.
The article is too long and I think there are many things that can be synthesised.
Things that can be synthesised. I would include a description of the material including a table and for more detail I would either add an appendix or upload it to github with the I would upload it to github along with the full description of the experiment.
Also, I think there are too many numerical tables that could be condensed into more representative figures that are already included in the paper.
The work is very interesting and represents a significant and novel work, although I would recommend at least a look at the previous comments regarding the structure.
On the other hand, I would recommend at least exploring how cryptographic algorithms influence cryptographic algorithms influence this consumption. Another limitation of IoT devices related to their reduced capabilities is security, as shown in "A Test Environment for Wireless Hacking in Domestic IoT Scenarios". IoT Scenarios'. I suggested exploring this as a potential future work.
Author Response
We sincerely appreciate the anonymous reviewer for his/her helpful and valuable comments for enriching and improving the quality and importance of the manuscript. The entire manuscript has been carefully revised according to reviewer’s comments. The most important changes to the revised manuscript are highlighted with a red color. Point-to-point responses to the specific questions and queries are presented below.
The main objective of this work is to study the impact of data compression on the power consumption of data compression on the power consumption of data transmission, using different data transmission, using different transmission modules, in a microcontroller-based system with very limited resources. The authors study the performance of different compression algorithms and transmission modules, taking into account the computational and memory complexity, as well as a performance analysis.
The proposal is to use an embedded device that transmits sensor data, based on the STM32F411 microcontroller. The STM32F411CE microcontroller is to be used as the transmitter module, and the nRF24L01+ board is to be used as the transmitter module, since it requires the least amount of memory. module, since it requires the least amount of power to transmit one byte of data. The authors conclude that the best option is to use the LZ78 algorithm in combination to achieve the highest energy and time efficiency.
In the case of image data, using the JPEG algorithm for compression gives the best results.
The article is too long and I think there are many things that can be synthesised.
Things that can be synthesised. I would include a description of the material including a table and for more detail I would either add an appendix or upload it to github with the I would upload it to github along with the full description of the experiment.
Also, I think there are too many numerical tables that could be condensed into more representative figures that are already included in the paper.
Response:
We thank the reviewer for this comment. We thoroughly analyzed the manuscript in order to shorten and to synthesis some elements. In more detail:
- Section 3.1 was synthesized, i.e., some descriptions were condensed or removed.
- Section 3.3 was synthesized, i.e., some descriptions were condensed or removed.
- Table 4 of the original manuscript was converted to a figure (Figure 2 in the revised manuscript).
- Table 6 of the original manuscript was converted to a figure (Figure 3 in the revised manuscript).
- Listing 7 and description of this listing was removed.
- Section 5 was synthesized, i.e., some descriptions were condensed or removed.
The work is very interesting and represents a significant and novel work, although I would recommend at least a look at the previous comments regarding the structure.
Response:
We thank the reviewer for this comment. As described above, the manuscript was modified to shorten and to synthesis some elements.
On the other hand, I would recommend at least exploring how cryptographic algorithms influence cryptographic algorithms influence this consumption. Another limitation of IoT devices related to their reduced capabilities is security, as shown in "A Test Environment for Wireless Hacking in Domestic IoT Scenarios". IoT Scenarios'. I suggested exploring this as a potential future work.
Response:
We thank the reviewer for this comment. We fully agree with the reviewer that cybersecurity is a serious threat to microcontroller-based system analyzed in our manuscript. We added in Conclusions this topic as a potential future direction for our research.
Round 2
Reviewer 1 Report
Comments and Suggestions for Authors
Many thanks for the efforts of the authors. My concerns have been addressed. I have no further comments.